# Energy System Contributions and Physical Activity in Specific Age Groups during Exergames

**DOI:** 10.3390/ijerph17134905

**Published:** 2020-07-07

**Authors:** Seung-Bo Park, Minjun Kim, Eunseok Lee, Doowon Lee, Seong Jun Son, Junggi Hong, Woo-Hwi Yang

**Affiliations:** Graduate School of Sports Medicine, CHA University, Seongnam-si, Gyeonggi-do 13503, Korea; parks0524@naver.com (S.-B.P.); minjunkim0113@gmail.com (M.K.); Eunseok4598@naver.com (E.L.); doowonlee1992@gmail.com (D.L.); seongjunson@gmail.com (S.J.S.); ptlhong@chauniv.ac.kr (J.H.)

**Keywords:** exergames, energy system contributions, oxygen uptake, lactate, sedentary behavior

## Abstract

Exergames have been recommended as alternative ways to increase the health benefits of physical exercise. However, energy system contributions (phosphagen, glycolytic, and oxidative) of exergames in specific age groups remain unclear. The purpose of this study was to investigate the contributions of three energy systems and metabolic profiles in specific age groups during exergames. Seventy-four healthy males and females participated in this study (older adults, *n* = 26: Age of 75.4 ± 4.4 years, body mass of 59.4 ± 8.7 kg, height of 157.2 ± 8.6 cm; adults, *n* = 24: Age of 27.8 ± 3.3 years, body mass of 73.4 ± 17.8 kg, height of 170.9 ± 11.9 cm; and adolescents, *n* = 24: Age of 14 ± 0.8 years, body mass of 71.3 ± 11.5 kg, height of 173.3 ± 5.2 cm). To evaluate the demands of different energy systems, all participants engaged in exergames named Action-Racing. Exergames protocol comprised whole-body exercises such as standing, sitting, stopping, jumping, and arm swinging. During exergames, mean heart rate (HR_mean_), peak heart rate (HR_peak_), mean oxygen uptake (VO_2mean_), peak oxygen uptake (VO_2peak_), peak lactate (Peak La^−^), difference in lactate (ΔLa^−^), phosphagen (W_PCr_), glycolytic (W_La_^−^), oxidative (W_AER_), and total energy demands (W_Total_) were analyzed. The contribution of the oxidative energy system was higher than that of the phosphagen or glycolytic energy system (65.9 ± 12% vs. 29.5 ± 11.1% or 4.6 ± 3.3%, both *p* < 0.001). The contributions of the total energy demands and oxidative system in older adults were significantly lower than those in adults and adolescents (72.1 ± 28 kJ, *p* = 0.028; 70.3 ± 24.1 kJ, *p* = 0.024, respectively). The oxidative energy system was predominantly used for exergames applied in the current study. In addition, total metabolic work in older adults was lower than that in adolescents and adults. This was due to a decrease in the oxidative energy system. For future studies, quantification of intensity and volume is needed to optimize exergames. Such an approach plays a crucial role in encouraging physical activity in limited spaces.

## 1. Introduction

The World Health Organization (WHO) has estimated that there will be a 10% reduction in physical inactivity by 2025. However, 28% of adults showed an increase in physical inactivity from 2001 to 2016 [1]. Physical inactivity and sedentary lifestyles are major factors for an increase in non-communicable diseases such as cardiovascular diseases and diabetes [2,3]. It has been documented that physical activity and mortality are strongly associated with each other, having an inverse dose–response relationship [4]. In 2017, 2.2 billion people worldwide spent many hours performing sedentary behaviors such as playing video games (mobile games, web-based games, computer games, and home entertainment games) [5].

While traditional video games can be played from a seated position, exergames (a portmanteau of “exercise” and “gaming”) comprise whole-body exercise in the form of running, dancing, jumping, and cycling, thereby enhancing physical activity [6,7,8]. Exergames enable improved exercise performance of the game player through immediate visual and auditory feedback such as enjoyment, motivation, and engagement [9]. Therefore, it has been hypothesized that playing exergames could be one potential strategy to increase physical activity levels and attenuate sedentary behaviors related to non-communicable diseases [10,11,12].

Previous studies have reported that exergames can increase energy expenditure (EE) in metabolic equivalents (METs) ranging from 1.5 to 6 METs [7] and improve maximal oxygen uptake (VO_2max_) [13]. Playing exergames can induce higher mean oxygen uptake (VO_2mean_), mean heart rate (HR_mean_), and maximal heart rate (HR_max_) than playing sedentary video games [14]. Graves et al. [15] have compared VO_2_, EE, and HR in three different age groups (adolescents, adults, and older adults) while playing Wii Fit and suggested its compliance with daily physical activity recommendations for all age groups. A systematic review has analyzed EE, HR, VO_2_, VO_2max_, METs, body composition, and cardiorespiratory fitness while playing exergames [13]. However, such approaches may lead to misleading conclusions concerning the energy system contributions of exergames known to enhance physical activity due to the different analyzing method of metabolic works.

Obtaining accurate metabolic profiles of specific age groups could play an important role in providing basic data for the development of an optimized program of exergames. However, the way in which different energy systems such as oxidative, phosphagen, and glycolytic systems actually contribute during exergames in specific age groups remains unclear. Therefore, the objective of this study was to investigate contributions of three energy systems and metabolic profiles in specific age groups during exergames. We hypothesized that energy contribution would be predominant in the oxidative system during exergames. We also hypothesized that exergames could lead to difference in total metabolic work in different age groups.

## 2. Materials and Methods

### 2.1. Participants

Seventy-six healthy male and female participants volunteered for this study (older adults, *n* = 26: Age of 75.4 ± 4.4 years, body mass of 59.4 ± 8.7 kg, height of 157.2 ± 8.6 cm; adults, *n* = 24: Age of 27.8 ± 3.3 years, body mass of 73.4 ± 17.8 kg, height of 170.9 ± 11.9 cm; and adolescents, *n* = 24: Age of 14 ± 0.8 years, body mass of 71.3 ± 11.5 kg, height of 173.3 ± 5.2 cm) (Table 1). Of those 76 participants, two dropped out. A total of 74 subjects completed this study. All participants were assessed with a medical examination report and a physical activity readiness questionnaire (PAR-Q). Inclusion criteria used in this study were as follows: (1) Healthy male and female participants (age: 13 to 83 years); (2) those who had sedentary lifestyles (participants engaged in only 30 min per week of physical activity); (3) those who had no history of cardiovascular, respiratory, or musculoskeletal system disease; and (4) those who were free of medication (all candidates had a medical interview and a 12-lead resting EKG examination to confirm eligibility by a doctor). Participants were educated to prohibit medications (6 h), caffeine (6 h), alcohol (48 h), and intense exercise (48 h) before every visit. In addition, they were informed about the overall experimental procedure. However, the purpose or hypotheses of this study were not explained to them. Written informed consent was obtained from all participants. This study was approved by the Institutional Review Board of CHA University (NO. 1044308-201910-HR-075-02).

### 2.2. Study Design

All participants visited the laboratory twice. During visit 1, participants were familiarized with exergames (Action-Racing, Cloud Gate, Inc., Gangnam, Seoul, Korea), which involved five activities (standing, sitting, stopping, jumping, and arm swinging) for 5 min (Figure 1). During visit 2, participants carried out the protocol of exergames comprising whole-body exercise by maintaining an exercise score of over 70 points (the achieved level of exercise intensity and movement score).

### 2.3. Exergames Procedures

In this study, participants completed a warm-up of running and stretching for 10 min. Before exergames, participants underwent five minutes of passive recovery in standing position to measure resting cardiorespiratory responses. They then performed exergames for five minutes. Exergames comprising five activity types were carried out in an area measuring 3.9 m × 2.8 m × 3.5 m with a screen measuring 3.9 m × 2.8 m in dimension. Gas data were recorded continuously until six minutes after the end of exergames. Blood lactate concentration was measured before exergames and every minute after exergames up to the seventh minute.

### 2.4. Physiological Measurements

Body composition was determined with a bioelectrical impedance analysis device (Inbody 720; Biospace Co. Ltd., Seoul, Korea). For trials of exergames, a capillary blood sample from the earlobe was subjected to blood lactate concentration measurement at rest and immediately at the end of exergames. Subsequently, a 20 μL blood sample was analyzed with an enzymatic–amperometric measuring method using a lactate analyzer (Biosen C-line, EKF diagnostic sales, GmbH, Magdeburg, Germany). Breath-by-breath analysis was performed to measure gas exchange using portable spirometry systems (Metamax 3B, Cortex Biophysik, Leipzig, Germany). Before exergames, a metabolism analyzer was calibrated using standard gas composition (15% O_2_, 5% CO_2_, Metamax 3B, Cortex Biophysik, Leipzig, Germany). A syringe was used to calibrate the flow and volume (Hans Rudolph Inc., Kansas, MO, USA). HR was continuously monitored with a Bluetooth^®^ heart rate transmitter strap (Polar H10, Polar Electro OY, Kempele, Finland) during exergames.

### 2.5. Calculating Energy System Contributions

To estimate energy demands of exergames, the contributions of oxidative, glycolytic, and phosphagen systems were estimated by measuring oxygen uptake (VO_2_) during activity, the delta of lactate (ΔLa^−^), and the fast component of excess post-exercise oxygen uptake after exergames (EPOC_FAST_), respectively [16]. Oxidative energy (W_AER_) was estimated by subtracting VO_2REST_ from VO_2_ area integrated overtime during exergames with a trapezoidal method [16]. Phosphagen system contribution (W_PCr_) was estimated considering EPOC_FAST_ after exergames. The fast component of mono-exponential models employed parameters to estimate W_PCr_. The slow component of a bi-exponential model was excluded [17]. The contribution of the glycolytic system (W_La_^−^) was calculated with lactate concentration (l mmol∙L^−1^ of La^−^ = 3 mL O_2_ kg^−1^ body mass) after exergames [18]. A caloric quotient of 20.92 kJ [19] was employed for all three different energy systems. The sum of the three energy systems indicated the total amount of energy contribution (W_TOTAL_). In addition, the contribution of the three energy systems was expressed as W_TOTAL_ percentage.

### 2.6. Statistical Analysis

Data were analyzed using GraphPad Prism 7.0 (GraphPad Software, La Jolla, CA, USA). Results of calculation of mean (M) and standard deviation (SD) are presented for all variables. In addition, 95% confidence interval (95% CI) was calculated for each sample. Normality of data distribution was performed using the Shapiro–Wilk test. After adjusting for the potential covariates (age, sex, and body mass), the analysis of specific age groups was assessed with the Kruskal–Wallis test when normality was not confirmed. Post-hoc statistically significant effects were determined using Dunn’s multiple comparison. Effect sizes (ES) eta squared (η^2^) were calculated for non-parametric tests. Based on the standard value of η^2^, effects were interpreted as follows: 0.01 ≤ η^2^ < 0.06, small effect; 0.06 ≤ η^2^ < 0.14, moderate effect; η^2^ ≥ 0.14, large effect [20]. When normality was confirmed for dependent variables, data were analyzed using one-way analysis of variance (ANOVA). Statistically significant effects were determined post-hoc using Tukey’s honestly significant difference (HSD) test. Statistical significance was set at *p* < 0.05. ES were evaluated using Cohen’s *f* value for parametric test and interpreted using thresholds proposed by Yigit and Mendes [21]: *f* < 0.1, small effect; *f* < 0.25, medium effect; and *f* < 4.0, large effect.

## 3. Results

### 3.1. Physiological Parameters

There was no significant difference between male and female between groups (*p* > 0.05). Physiological responses to exergames were examined. Results are shown in Table 1. No significant differences were observed between specific age groups for VO_2peak_ and ΔLa^−^ (F_(2,71)_ = 0.674, *p* = 0.513; *f* = 0.019, H_(2)_ = 0.632, *p* = 0.729; η^2^ = 0.47, respectively). However, HR_mean_, HR_peak_, Peak La^−^, and VO_2mean_ were significantly different between specific age groups (F_(2,71)_ = 8.221, *p* < 0.001; *f* = 0.232, F_(2,71)_ = 6.493, *p* = 0.003; *f* = 0.183, F_(2,71)_ = 3.482, *p* = 0.036; *f* = 0.098, F_(2,71)_ = 4.867, *p* = 0.01; *f* = 0.137, respectively). A post-hoc test revealed that the HR_mean_ of older adults was significantly lower than that of adults and adolescents (*p* < 0.001, 95% CI: −28.34 to −6.37; *p* = 0.007, 95% CI: −25.35 to −3.3, respectively). The HR_peak_ of older adults was also significantly lower than that of adults (*p* = 0.002, 95% CI: −29.07 to −5.61). The Peak La^−^ in older adults was significantly higher than that in adolescents (*p* = 0.027, 95% CI: 0.07 to 1.37). The VO_2mean_ of older adults was significantly lower than that of adolescents (*p* = 0.012, 95% CI: −5.08 to −0.53).

### 3.2. Energy System Contributions

There were statistically significant differences between energy system contributions under exergames (H_(2)_ = 171.162, *p* < 0.001; η^2^ = 0.628). Post-hoc tests revealed that the contribution of the oxidative energy system was higher than that of the phosphagen or glycolytic energy system [41.89 ± 13.92 kJ (65.9 ± 12%) vs. 20.54 ± 15.64 kJ (29.5 ± 11.1%) or 3.24 ± 3.32 kJ (4.6 ± 3.3%), both *p* < 0.001] (Figure 2 and Figure 3). There were statistically significant differences in W_TOTAL_ (H_(2)_ = 9.349, *p* = 0.009; η^2^ = 0.087) among specific age groups (Table 1). Post-hoc tests indicated that W_TOTAL_ in older adults was significantly lower (55.4 ± 23.2 kJ) than in adults and adolescents (72.1 ± 28 kJ, *p* = 0.028; and 70.3 ± 24.1 kJ, *p* = 0.024). There were also statistically significant differences in oxidative system contribution among specific age groups (H_(2)_ = 23.901, *p* < 0.001; η^2^ = 0.292). Post-hoc tests revealed that the contribution of the oxidative system in older adults 31.7 ± 10.7 kJ (58.9 ± 9.7%) was significantly lower than that in adults and adolescents (47.9 ± 15.6 kJ (67.7 ± 7.5%), *p* < 0.001; and 46.9 ± 8.2 kJ (70.8 ± 14.4%), *p* < 0.001) (Figure 2 and Figure 3).

## 4. Discussion

Information about different metabolic demands of exergames plays an essential role in the future development of a game protocol to optimize indoor physical activity. In this context, analysis of energy system contribution toward exergames has not been previously reported. The present study is the first to estimate contributions of three energy systems in specific age groups during exergames. This study revealed that the oxidative energy system was predominantly utilized (65.9% of total energy demand) during exergames, and this was significantly lower in older adults than in adolescents. The present study also found that total metabolic work in older adults was lower than in adults and adolescents.

In the current study, contributions of phosphagen and glycolytic systems were estimated to be 29.5% and 4.6% of total metabolic cost, respectively. These results indicate that lactate produced during exergames might have been eliminated by the lactate shuttle mechanism (re-synthesis) [22,23,24]. Exergames used in this study comprised whole-body exercises such as standing, sitting, stopping, jumping, and arm swinging. These exercises had relatively moderate intensity. Such an intensity of exercise inevitably leads to a predominant contribution of the oxidative energy system compared to phosphagen and glycolytic systems [7,16]. Campos et al. [16] have reported that energy contributions of phosphagen, glycolytic, and oxidative energy systems in taekwondo athletes are 30%, 4%, and 66% in relative values, respectively. These results are consistent with our findings of energy system contributions during exergames.

Outcomes of our study indicated that W_TOTAL_ (kJ) of older adults was lower than that of adults or adolescents during exergames (Figure 3). These results are similar to results of previous studies, showing that older adults have lower EE than adolescents and young adults [15]. During exergames, differences in W_TOTAL_ (kJ) between different groups seem to be caused by a lower oxidative energy demand in older adults. In the current study, the oxidative system contribution in older adults (58.9%) was lower than that in adults (67.7%) or adolescents (70.8%) (Figure 2). Older adults might have more difficulty moving than adults and adolescents. The lower W_TOTAL_ (kJ) in older adults stemmed from physical limitations, which affect their mobility and neural control [9,15]. Another important factor that might contribute to difference in W_TOTAL_ (kJ) was that older adults weighed about 24% less than adults and adolescents.

A previous study has reported that enjoyment factors in exergames can increase the level of physical fitness and demonstrated that outcomes of exergames are as effective as traditional exercise [25]. Given these positive effects, analyzing demands of energy systems during exergames is essential to the development of game protocol in which exercise intensity and volume can be systematically established. Such an approach could lead to improvement of physical fitness in specific groups such as older adults. Therefore, we believe that the importance of indoor exercise is likely to increase further and that there will a novel paradigm for types of physical activity. This implies that exercise programs that can be performed in constrained spaces should be increased. In particular, exergames present a new exercise model to alleviate the challenges of specific groups who have limited physical activity due to their narrow activity radius and space restrictions.

Our study has some limitations. First, results of this study cannot be generalized to various types of exergames for promoting physical activity. In addition, participants in each age group were unable to complete all mission tasks required for playing exergames. Although older adults tried to follow the protocol of exergames and achieved an exercise protocol score, movements of older adults were fewer than those of adults and adolescents.

## 5. Conclusions

For the exergames applied in the current study, the oxidative energy system was predominantly used. This study revealed that total metabolic work in older adults was lower than in adults or adolescents. This was related to a lower oxidative energy system in older adults because of their fewer exercise movements. In future studies, intensity and volume should be quantified to optimize programs of exergames. Such an approach of performing exergames plays a crucial role in encouraging physical activity in limited spaces. Such an approach could also be applied to alleviate sedentary behavior-induced clinical symptoms such as cardiovascular and metabolic diseases that are related to the oxidative energy system.

## Figures and Tables

**Figure 1 ijerph-17-04905-f001:**
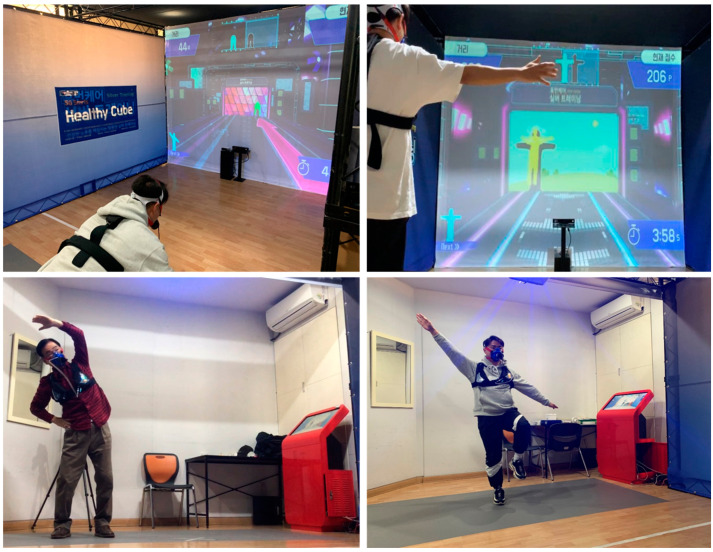
Exergames (Action-Racing).

**Figure 2 ijerph-17-04905-f002:**
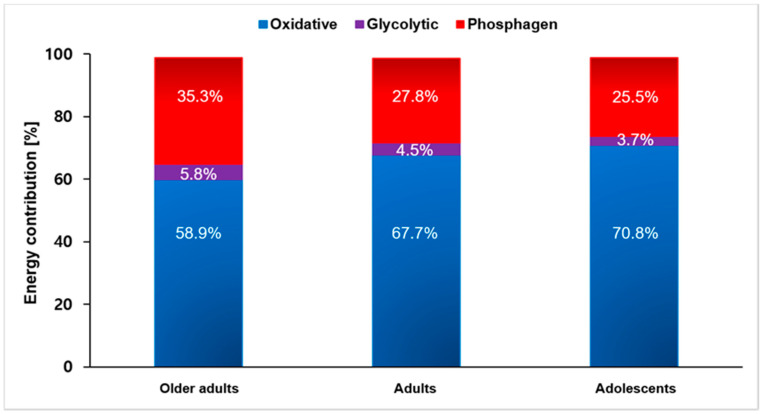
Average contribution of each energy system during exergames in percentage (%).

**Figure 3 ijerph-17-04905-f003:**
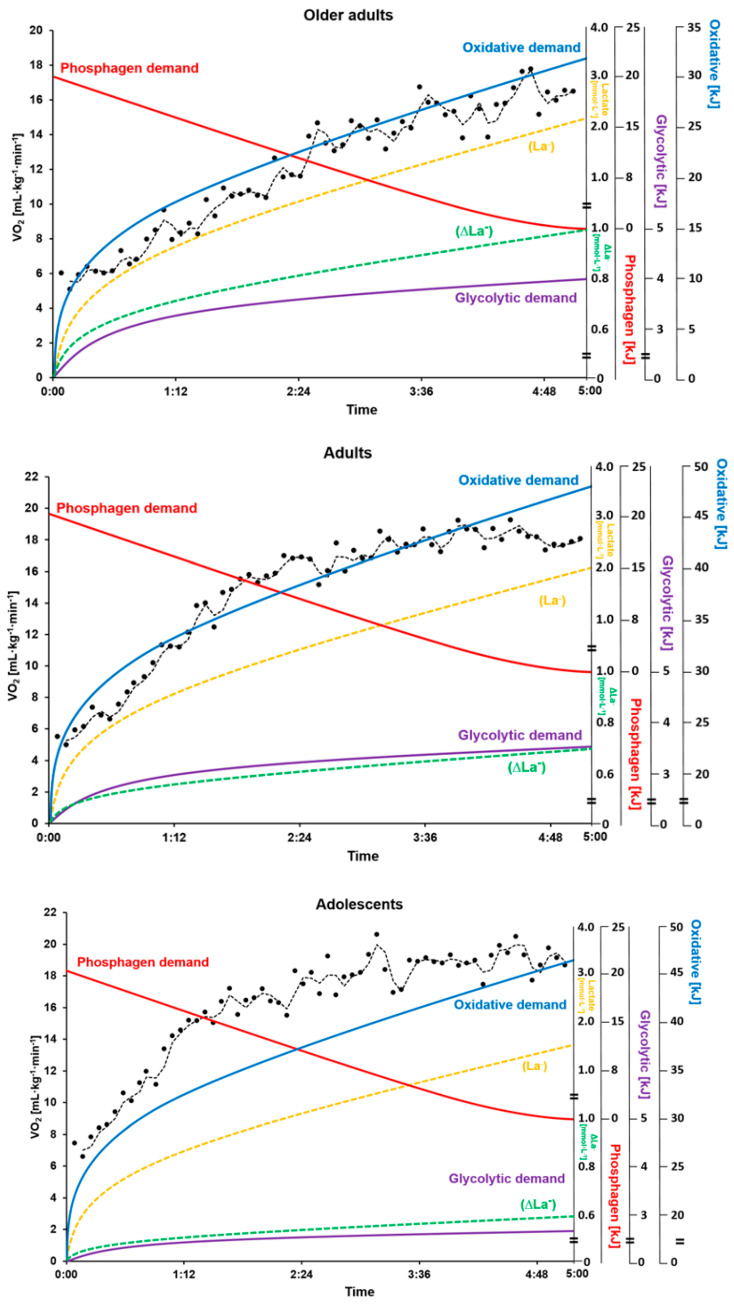
Mean contribution of each energy system during exergames in absolute values (kJ).

**Table 1 ijerph-17-04905-t001:** Physiological responses and calculated energy system contributions during exergames.

	Older Adults (*n* = 26)	Adults (*n* = 24)	Adolescents (*n* = 24)
Age (years)	75.4 ± 4.4	27.8 ± 3.3	14 ± 0.8
Body mass (kg)	59.4 ± 8.7	73.4 ± 17.8	71.3 ± 11.5
Height (cm)	157.2 ± 8.6	170.9 ± 11.9	173.3 ± 5.2
HR_mean_ (beats·min^−1^)	105.3 ± 18.8	122.8 ± 15.9 ^a,^***	119.7 ± 13.4 ^a,^**
HR_peak_ (beats·min^−1^)	115.1 ± 19.6	132.4 ± 18.4 ^a,^*	126.3 ± 12.9
VO_2mean_ (mL·kg^−1^·min^−1^)	14.1 ± 3.7	16.3 ± 3.1	16.9 ± 3.1 ^a^ *
VO_2peak_ (mL·kg^−1^·min^−1^)	19.2 ± 4.9	19.9 ± 3.8	20.6 ± 4.1
Peak La^−^ (mmol·L^−1^)	2.3 ± 1.3	2.0 ± 0.9	1.6 ± 0.4 ^a^ *
∆La^−^ (mmol·L^−1^)	1.0 ± 1.1	0.7 ± 6.0	0.6 ± 0.3
W_PCR_ (kJ)	20.0 ± 10.9	20.7 ± 11.5	20.9 ± 22.7
W_PCR_ (%)	35.3 ± 7.7	27.8 ± 7.5	25.5 ± 14.5 ^a^ ***
W_La_^−^ (kJ)	3.7 ± 4.1	3.5 ± 3.6	2.5 ± 1.4
W_La_^−^ (%)	5.8 ± 4.3	4.5 ± 2.7	3.7 ± 2.1
W_AER_ (kJ)	31.7 ± 10.7	47.9 ± 15.6 ^a^ ***	46.9 ± 8.2 ^a^ ***
W_AER_ (%)	58.9 ± 9.7	67.7 ± 7.5	70.8 ± 14.4 ^a^ ***
W_TOTAL_ (kJ)	55.4 ± 23.2	72.1 ± 28 ^a^ *	70.3 ± 24.1 ^a^ *

Data are present as mean ± S.D. HR_mean_: Mean heart rate; HR_peak_: Peak heart rate; VO_2mean_: Mean oxygen uptake; VO_2peak_: Highest oxygen uptake; Peak La^−^: Highest lactate concentration; ΔLa^−^: Difference of lactate concentration after the exergames at the beginning of the exergames; W_PCR_: Phosphagen system contribution; W_La_^−^: Glycolytic system contribution; W_AER_: Oxidative system contribution. a: Older adults. * *p* < 0.05; ** *p* < 0.01; *** *p* < 0.001.

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
