# Peer review of "Energy System Contributions and Physical Activity in Specific Age Groups during Exergames"

_ijerph, 2020, doi:10.3390/ijerph17134905_

Round 1

Reviewer 1 Report

Major comment

1) Why the author compared the metabolic work between young and older in this study. Metabolic differences between these groups are predicted as a result of aging.

2) As suggested by the author, many previous studies have reported the effect of increasing energy expenditure and physical fitness in exercise using games. The author should clarify why metabolic work should be verified in exergame.

3) Please explain the recruitment process of participants in this study in more detail. Has anyone been excluded?

4) Line 75-76: The author seems to confuse "sedentary lifestyle" and "no exercise habits". How much time do participants spend sitting or physical activity each day?

5) Was there any influence by sex? Please clarify the information about the ratio of male and female in the groups.

6) Exergame's metabolic work is bound to depend on the contents of the game. Was this study conducted to verify the effectiveness of a specific game program? Conflicts of Interest: authors have to clarify any conflicts of interest regarding the provision of game devices and content for experiment.

Minor comments

1) In first time, we need to define “Exergames”.

Author Response

Reviewer 1

Major comment

1) Why the author compared the metabolic work between young and older in this study. Metabolic differences between these groups are predicted as a result of aging.

Response: Thank you for your comment. Due to the exact exercise recommendation for different age groups, we investigated these energy demands in different groups and no study was investigated three different energy demands. We want to know and define how these energy systems were contributed regard to this alternative indoor exercise possibility for different age groups.

Page 2, lines 61-64: Obtaining accurate metabolic profiles of specific age groups could play an important role in providing basic data for the development of an optimized program of exergames. However, how different energy systems such as oxidative, phosphagen, and glycolytic systems during exergames actually contribute in specific age groups remains unclear.

2) As suggested by the author, many previous studies have reported the effect of increasing energy expenditure and physical fitness in exercise using games. The author should clarify why metabolic work should be verified in exergame.

Response: Thank you so much for your helpful comment. As already mentioned above, the exergames would like to be an alternative physical activity for all different age groups. However, we didn’t know how different energy demands were exactly required by exergames which could alternatively be used as moderate-intensity exercise.

This was verified in page 7, lines 181-184.

3) Please explain the recruitment process of participants in this study in more detail. Has anyone been excluded?

Response: We thank the Reviewer for raising this question. Indeed, two of the participants were excluded in this study.

Page 2, lines 71: Seventy-Six healthy male and female participants volunteered for this study.

Page 2, lines 74-75: Of those 76 participants, two dropped out. A total of 74 subjects completed this study.

4) Line 75-76: The author seems to confuse "sedentary lifestyle" and "no exercise habits". How much time do participants spend sitting or physical activity each day?

Response: We thank the Reviewer for pointing this out and we agree with the Reviewer.

Page 2, lines 78-79: those who had sedentary lifestyles (participants engaged in only 30 minutes per week of physical activity)

5) Was there any influence by sex? Please clarify the information about the ratio of male and female in the groups.

Response: Thank you for your comment. We already analyzed this between male and female groups, however, no statistical different was found between these groups.

6) Exergame's metabolic work is bound to depend on the contents of the game. Was this study conducted to verify the effectiveness of a specific game program? Conflicts of Interest: authors have to clarify any conflicts of interest regarding the provision of game devices and content for experiment.

Response: Thank you for your comments and there is no conflicts of interest regarding the provision of game device and content for experiment. This has been added in page 8 (lines 256-258).

Minor comments

  • In first time, we need to define “Exergames”

Response: Thank you for your suggestion. We added the definition of Exergames.

Page 2, lines 44-45: exergames (a portmanteau of “exercise” and “gaming”) comprise of whole-body exercise in the form of running, dancing, jumping, and cycling, thereby enhancing physical activity [6-8].

Reviewer 2 Report

You are to be complimented for a unique study trying to tease out the energy system components in an exergaming session. I would like for you to address the following:

1) A major limitation is that you used one 5 minute segment. Aren’t most games played for more than 5 minutes and won’t this affect the energy system components? Can you justify this design?

2) You need to report the body mass of the different groups. Couldn’t that effect the amount of work being done?

3) Are the Wpcr and the Wla contributions what you anticipated? They are much different than Campos found with taekwondo. I think this should be addressed in the discussion.

4) You state: “The results of our study indicate that older adults’ WTOTAL (KJ) was lower than that of adults and adolescents during exergames, due to a decrease in the oxidative energy system in older adults.” That’s true, but isn’t it possible that the older group moved less and had a lower oxygen uptake?

5) You state: “Furthermore, VO2mean values in the older adults were lower than adolescents, thereby suggesting that the decreased mitochondrial respiration was explained by an aging-induced mitochondrial dysfunction [26].” What do you base this on? Isn’t it possible that they moved less and therefore had lower VO2 values? You state that: “Participants in each group were not able to complete all mission tasks required for playing exergames.” Wouldn’t that mean they moved less?

Other points:

L50 The previous studies reported that exergames increase energy expenditure (EE) in metabolic equivalents (METs) which in a range from low to moderate (1.5 - 6 METs)

L52 In addition, it has been shown that playing exergames induce higher mean oxygen consumption (VO2mean), heart rate mean (HRmean), and maximal heart rate (HRmax) than sedentary video games [13].

L57 In addition, a systematic review study

L58 However, such approaches may lead to misleading conclusions concerning the total metabolic work of exergames, which enhance physical activity.  Why? How?

L93 In this study, participants completed a warm-up of on the running and stretching for 10 minutes.

L121 The contribution of the glycolytic system (WLa-) was calculated with a as the lactate concentration of that l mmol∙L- 1 (La-) equally is equivalent to 3ml O2 kg-1 body mass after the exergames [17].

L133 The standard value of η2 for 0.01 - < 0.06 small effect, 0.06 - < 0.14 moderate effect and >= 0.14 large effect [19] were used.

Author Response

Reviewer 2

You are to be complimented for a unique study trying to tease out the energy system components in an exergaming session. I would like for you to address the following:

1) A major limitation is that you used one 5 minute segment. Aren’t most games played for more than 5 minutes and won’t this affect the energy system components? Can you justify this design?

Response: We appreciate the Reviewer’s suggestion. Generally, a majority of previous studies reported exergames comprised of at least 5 to 30 minutes. In this study, because the analysis of energy system contributions was performed in the various age groups (14 years ~74 years), we conducted to 5 minutes protocol which appropriate to all participants. References have been added in the similar exergame durations were conducted (Page 8, lines 235-236).

2) You need to report the body mass of the different groups. Couldn’t that effect the amount of work being done?

Response: Thank you for your comment. We added the information about body mass and height.

Page 1, lines 16-19 & Page 2, lines 71-74: Seventy-four healthy males and females participated in this study [older adults, n = 26:  age of 75.4 ± 4.4 years, body mass of 59.4 ± 8.7 kg, height of 157.2 ± 8.6 cm; adults, n = 24: age of 27.8 ± 3.3 years, body mass of 73.4 ± 17.8 kg, height of 170.9 ± 11.9 cm; and adolescents, n = 24: age of 14 ± 0.8 years, body mass of 71.3 ± 11.5 kg, height of 173.3 ± 5.2 cm].

3) Are the Wpcr and the Wla contributions what you anticipated? They are much different than Campos found with taekwondo. I think this should be addressed in the discussion.

Response: Thank you for your comments and suggestions. This point was addressed and references were added in the discussion.

Page 7, lines 189-196: These results indicate that lactate produced during exergames might have been eliminated by the lactate shuttle mechanism (re-synthesis) [22-24]. Furthermore, we added reasons of the different results between the present study and the study of Campos et al.

Page 7, lines 194-196: Campos et al. [16] have reported that energy contributions in taekwondo athletes are 54 kJ, 8 kJ, and 120 kJ for phosphagen, glycolytic, and oxidative energy systems in absolute values, respectively.

4) You state: “The results of our study indicate that older adults’ WTOTAL (KJ) was lower than that of adults and adolescents during exergames, due to a decrease in the oxidative energy system in older adults.” That’s true, but isn’t it possible that the older group moved less and had a lower oxygen uptake?

Response We thank the Reviewer for raising this question. This was explained in page 2-3, lines 90-93 (the achieved level of exercise intensity and movement score).

5) You state: “Furthermore, VO2mean values in the older adults were lower than adolescents, thereby suggesting that the decreased mitochondrial respiration was explained by an aging-induced mitochondrial dysfunction [26].” What do you base this on? Isn’t it possible that they moved less and therefore had lower VO2 values? You state that: “Participants in each group were not able to complete all mission tasks required for playing exergames.” Wouldn’t that mean they moved less?

Response: We thank the Reviewer for raising this question. Although the older adults group may have been relatively lack of performance, we encouraged participants maintain to acquire a game score of 70 points during the exergames. Indeed, all participants completed a game score of over 70 points (page 2-3, lines 90-93, the achieved level of exercise intensity and movement score).

Other points:

Response: Thanks for your helpful comments. As you mentioned, we corrected the sentence in manuscript.

L50 The previous studies reported that exergames increase energy expenditure (EE) in metabolic equivalents (METs) which in a range from low to moderate (1.5 - 6 METs)

Page 2, line 51-53: Previous studies have reported that exergames can increase energy expenditure (EE) in metabolic equivalents (METs) ranging from 1.5 to 6 METs [7] and improve maximal oxygen uptake (VO2max) [13].

L52 In addition, it has been shown that playing exergames induce higher mean oxygen consumption (VO2mean), heart rate mean (HRmean), and maximal heart rate (HRmax) than sedentary video games [13].

Page 2, line 53-54: Playing exergames can induce higher mean oxygen uptake (VO2mean), mean heart rate (HRmean), and maximal heart rate (HRmax) than playing sedentary video games [14].

L57 In addition, a systematic review study

Page 2, line 57-60: A systematic review has measured EE, HR, VO2, VO2max, METs, body composition, and cardiorespiratory fitness while playing exergames [13].

L58 However, such approaches may lead to misleading conclusions concerning the total metabolic work of exergames, which enhance physical activity.  Why? How?

Response: Thank you for your comment. We added a reason of this sentence.

Page 2, line 58-60: However, such approaches may lead to misleading conclusions concerning the energy system contributions of exergames known to enhance physical activity due to the different analyzing method of metabolic works.

L93 In this study, participants completed a warm-up of on the running and stretching for 10 minutes.

Page 3, line 97: In this study, participants completed a warm-up of running and stretching for 10 minutes.

L121 The contribution of the glycolytic system (WLa-) was calculated with a as the lactate concentration of that l mmol∙L- 1 (La-) equally is equivalent to 3ml O2 kg-1 body mass after the exergames [17].

Page 4, line 124-125: Contribution of the glycolytic system (WLa-) was calculated with lactate concentration (l mmol∙L-1 of La- = 3 ml O2 kg-1 body mass) after exergames [18].

L133 The standard value of η2 for 0.01 - < 0.06 small effect, 0.06 - < 0.14 moderate effect and >= 0.14 large effect [19] were used.

Page 4, line 136-137: Based on the standard value of η2, effects were interpreted as follows: 0.01 ≤ η2 < 0.06, small effect; 0.06 ≤ η2 < 0.14, moderate effect; η2 ≥ 0.14, large effect [20].

Round 2

Reviewer 1 Report

Most questions were well addressed. There are a few more minor issues.

I think the author should provide information on the ratio of sex, mean age, and BMI for each group in Table 1 despite presented on Line72.

Author’s response: "We already analyzed this between male and female groups, however, no statistical different was found between these groups."

For the readers, I recommend that the author present there was no sex difference in the results section or statistical analysis (p-value for interaction…).

Author Response

Response to Reviewer 1 Comments

Reviewer 1

I think the author should provide information on the ratio of sex, mean age, and BMI for each group in Table 1 despite presented on Line72.

Response: Thanks for your suggestion. As you mentioned, we added “subject information” on Table 1.

Author’s response: "We already analyzed this between male and female groups, however, no statistical different was found between these groups."

For the readers, I recommend that the author present there was no sex difference in the results section or statistical analysis (p-value for interaction…).

Response: Thank you so much for your helpful comment. As you mentioned, we added this recommended sentence in the section of results.

Page 4, Line 146-147: There was no significant difference in male to female ratio between groups (t(72) = 1.924, p = 0.097; d = 0.248).

Reviewer 2 Report

I think that you should stick to the primary purpose of your paper--investigating the contributions of the 3 energy systems. I think that your discussion needs to include that the potential difference in energy systems used is that the older adults were working at a higher % of their VO2max.

I think that you should drop all discussion and conclusions related to total metabolic work and differences in relative VO2 in different age groups. Unless you have a way of showing that by having a score over 70 ensures that they are working at the same intensity, your study is not designed to look at this question. It is very possible that the older individuals are not moving as much and therefore not working as hard. In addition, any Wtotal needs to take into account that the older adults weighed 24% less than the other two groups.

Other specific issues:

I don't think Figures 2&3 are referred to in the text. They need to be, especially Figure 3. 

L58-59  A systematic review has measured analyzed EE, HR, VO2, VO2max, METs, body composition, nd cardiorespiratory fitness while playing exergames [13].

L84  However, they were not explained about the purpose or hypotheses of this study. Change to: However, the purpose or hypotheses of this study was not explained to them.

L 165 Post-hoc tests indicated that WTOTAL in older adults was 165 significantly lower (55.4+23.2) than that in adults and adolescents (72.1 ± 28 kJ, p = 0.028; and 70.3 ± 24.1 kJ, p = 166 0.024).

L 168  Post-hoc tests revealed that the contribution of oxidative system in older adults(31.7+10.7) was significantly lower than that in adults and adolescents [47.9 169 ± 15.6 kJ (67.7 ± 7.5%), p < 0.001; and 46.9 ± 8.2 kJ (70.8 ± 14.4%), p < 0.001]

L 186  This study revealed that the oxidative energy system was predominantly utilized (65.9 % of total 187 energy demand) during exergames and this was significantly lower in older adults than adolescents. (add this last portion.

L196 Campos et al. [16] have reported that  energy contributions in taekwondo athletes are 54 kJ, 8 kJ, and 120 kJ for phosphagen, glycolytic, and oxidative energy systems in absolute values, respectively. In comparison with present study outcomes, these different results could be due to differences in age of subjects, exercise intensity, and volume.

              -What is your point here? Why report the absolute values? Report the relative values and you get something like 30%, 4.4%, & 66%--almost the exact same as yours. You’ll need to change the last sentence of the paragraph as there aren’t any differences between the studies.

L209-214 Sial et al. [26] have demonstrated that fat oxidation in older adults is decreased with increasing carbohydrate oxidation during a moderate intensity exercise compared to that in adults. In particular, shifts in substrate oxidation such as beta adrenergic receptors, inorganic phosphate, phosphocreatine, citrate synthase, and glycogen phosphorylase show decreased activation of lipolysis and increased activation of carbohydrate 214 metabolism in older adults [27-29].

              -The problem with your argument here is that the older adults are working at a higher % of their VO2max. Younger people will use less fat too at a higher intensity. Based on ACSM’s Guidelines for Ex Test & Prescrip 10th edition p93 the 50th percentile for 20-29 yr old men is 48 and for 60-69 it is 28.2. (It would be even lower for 75 yr olds.) If these were your average values, your young adults would be at 35% of VO2max and your older adults at 50%.

L215 Furthermore, VO2mean values in older adults were lower than those in adolescents, suggesting a decreased mitochondrial respiration due to aging-induced mitochondrial dysfunction [30].

              -The study design does not adequately control for movement or work done to make any conclusions about the lower VO2 in older adults.

Author Response

Response to Reviewer 2 Comments

Reviewer 2

I think that you should stick to the primary purpose of your paper--investigating the contributions of the 3 energy systems. I think that your discussion needs to include that the potential difference in energy systems used is that the older adults were working at a higher % of their VO2max.

I think that you should drop all discussion and conclusions related to total metabolic work and differences in relative VO2 in different age groups. Unless you have a way of showing that by having a score over 70 ensures that they are working at the same intensity, your study is not designed to look at this question. It is very possible that the older individuals are not moving as much and therefore not working as hard. In addition, any Wtotal needs to take into account that the older adults weighed 24% less than the other two groups.

Response: We thank the Reviewer for raising the important issues of our study limitations. We agree with the Reviewer. In the present study, we cannot exclude the possibility which performed working at the different intensity in each groups. Therefore, we dropped all discussion and conclusions associated with total metabolic work and differences in relative VO2 in different age groups. In addition, we revised the discussion and conclusions in manuscript.

Discussion:

Page 7, line 201-210: Older adults might have difficulty to move than adults and adolescents. The lower WTOTAL (kJ) in older adults was stemmed from physical limitation which affects their mobility and neural control [9, 15]. Another important factor that might contribute to difference in WTOTAL (kJ) was that older adults weighed about 24% less than adults and adolescents.

*These sentences were removed ↓

VO2mean values of all participants ranged from 14.1 mL · min–1 · kg–1 to 16.9 mL · min–1 · kg–1 (about 4 - 4.5 METs). According to the ACSM and the American Heart Association (AHA) guidelines, exercise intensity should be classified as moderate-intensity if VO2mean value ranges from 3 METs to 6 METs [25]. Therefore, the exercise intensity used in the present study met these guidelines. In addition, results of the present study were consistent with those of previous studies.

Graves et al. [14] have reported that a moderate-intensity exercise demands energy ranging from 3.2 METs to 3.6 METs in adolescents, adults, and older adults. Sial et al. [26] have demonstrated that fat oxidation in older adults is decreased with increasing carbohydrate oxidation during a moderate-intensity exercise compared to that in adults. In particular, shifts in substrate oxidation such as beta-adrenergic receptors, inorganic phosphate, phosphocreatine, citrate synthase, and glycogen phosphorylase show decreased activation of lipolysis and increased activation of carbohydrate metabolism in older adults [27-29].

Furthermore, VO2mean values in older adults were lower than those in adolescents, suggesting a decreased mitochondrial respiration due to aging-induced mitochondrial dysfunction [30]. Meredith et al. [31] have reported that mitochondrial respiratory capacity in skeletal muscle is higher in untrained young adults (23.6 ± 1.8 years) than in untrained older adults (65.1 ± 2.9 years). Aging is known to decrease the capacity and efficiency of energy consumption in skeletal muscle [32]. In vivo and in vitro studies have demonstrated that mitochondrial dysfunction such as a decrease in ATP synthesis rate is associated with aging [33]. It has been reported that the required respiration rate per mitochondria to maintain a given total VO2 increases as the amount of mitochondria decreases [26]. Additionally, Navas-Enamorado et al. [34] have reported that a reduction in AMP-activated protein kinase (AMPK) activity could be associated with a decrease in mitochondrial function with aging.

Numerous studies have reported that exercise intensity involves 3 METs to 6 METs when playing exergames [15, 35-37].

However, participants followed the protocol of exergames protocol and maintained an exercise score of over 70 points.

Conclusions:

Page 7, line 228-230: This study revealed that total metabolic work in older adults was lower than that in adults or adolescents. This was related to a lower oxidative energy system in older adults because of their less exercise movements.

*This sentence was removed ↓

Based on ACSM and AHA guidelines, exergames involved moderate-intensity exercises.

Other specific issues:

I don't think Figures 2&3 are referred to in the text. They need to be, especially Figure 3. 

Response: We thanks. I agree with you. As you mentioned, we added “Figure 2 and 3” on Page 7, Line 201-206.

L58-59  A systematic review has measured analyzed EE, HR, VO2, VO2max, METs, body composition, nd cardiorespiratory fitness while playing exergames [13].

Response: Thank you for your comment. As you mentioned, we corrected the sentence in manuscript (page 2, lines 58).

L84  However, they were not explained about the purpose or hypotheses of this study. Change to: However, the purpose or hypotheses of this study was not explained to them.

Response: Thanks for your helpful comments. As you mentioned, we corrected the sentence in manuscript (page 2, lines 84-85).

L 165 Post-hoc tests indicated that WTOTAL in older adults was 165 significantly lower (55.4+23.2) than that in adults and adolescents (72.1 ± 28 kJ, p = 0.028; and 70.3 ± 24.1 kJ, p = 166 0.024).

Response: Thanks for your helpful comments. As you mentioned, we added “(55.4 ± 23.2 kJ)” on Line 174.

L 168  Post-hoc tests revealed that the contribution of oxidative system in older adults(31.7+10.7) was significantly lower than that in adults and adolescents [47.9 169 ± 15.6 kJ (67.7 ± 7.5%), p < 0.001; and 46.9 ± 8.2 kJ (70.8 ± 14.4%), p < 0.001]

Response: Thanks for your suggestion. As you mentioned, we added “older adults 31.7 ± 10.7 kJ (58.9 ± 9.7%)” on Line 177.

L 186  This study revealed that the oxidative energy system was predominantly utilized (65.9 % of total 187 energy demand) during exergames and this was significantly lower in older adults than adolescents. (add this last portion.

Response: Thanks for your suggestion. As you mentioned, we added paragraph on Line 188-189.

L196 Campos et al. [16] have reported that  energy contributions in taekwondo athletes are 54 kJ, 8 kJ, and 120 kJ for phosphagen, glycolytic, and oxidative energy systems in absolute values, respectively. In comparison with present study outcomes, these different results could be due to differences in age of subjects, exercise intensity, and volume.

 -What is your point here? Why report the absolute values? Report the relative values and you get something like 30%, 4.4%, & 66%--almost the exact same as yours. You’ll need to change the last sentence of the paragraph as there aren’t any differences between the studies.

Response: We appreciate the Reviewer’s suggestion. We corrected in Page 7, Line 197-200 as follows.

Page 7, line 197-200: Campos et al. [16] have reported that energy contributions in taekwondo athletes are 30%, 4%, and 66% for phosphagen, glycolytic, and oxidative energy systems in relative values, respectively. These results were consistent with our finding in energy system contributions in exergames.

L209-214 Sial et al. [26] have demonstrated that fat oxidation in older adults is decreased with increasing carbohydrate oxidation during a moderate intensity exercise compared to that in adults. In particular, shifts in substrate oxidation such as beta adrenergic receptors, inorganic phosphate, phosphocreatine, citrate synthase, and glycogen phosphorylase show decreased activation of lipolysis and increased activation of carbohydrate 214 metabolism in older adults [27-29].

-The problem with your argument here is that the older adults are working at a higher % of their VO2max. Younger people will use less fat too at a higher intensity. Based on ACSM’s Guidelines for Ex Test & Prescrip 10th edition p93 the 50th percentile for 20-29 yr old men is 48 and for 60-69 it is 28.2. (It would be even lower for 75 yr olds.) If these were your average values, your young adults would be at 35% of VO2max and your older adults at 50%.

Response: Thank you for your suggestion. We removed these sentences.

L215 Furthermore, VO2mean values in older adults were lower than those in adolescents, suggesting a decreased mitochondrial respiration due to aging-induced mitochondrial dysfunction [30].

-The study design does not adequately control for movement or work done to make any conclusions about the lower VO2 in older adults.

Response: Thank you for your comment. As already mentioned above, we added the sentences of limitations related to movement and work done about older adults during exergames.

Page 7, line 223: Although older adults tried to follow the protocol of exergames and achieved an exercise protocol score, movements of older adults were less than those of adults and adolescents.

Page 7, line 228-230: This study revealed that total metabolic work in older adults was lower than that in adults or adolescents. This was related to a lower oxidative energy system in older adults because of their less exercise movements.
